# Investigating water movements around a shallow shipwreck in Big Tub Harbour of Lake Huron: Implications for managing and preserving underwater shipwrecks

**Bryan Flood** [1‡]*, **Lakshika Girihagama** [1‡], **Mathew G. Wells** [1], **Reza Valipour** [2], **Patricia Semcesen** [1], **Scott Parker** [3]

**1** University of Toronto Scarborough, Toronto, ON, Canada, **2** Environment and Climate Change Canada, Burlington, ON, Canada, **3** Parks Canada, Tobermory, ON, Canada

‡ BF and LG are co-first authors on this work.
* bryan.flood@mail.utoronto.ca

## Abstract

The Sweepstakes, in Fathom Five National Marine Park, is Ontario's most iconic shipwreck with over 100,000 visitors each summer. Continued exposure to water currents has directly and indirectly affected the integrity of the wreck and resulted in management interventions including efforts to stabilize the wreck and control vessel activity (both duration and speed). Despite these efforts, a scour ring is present in the sediment around the Sweepstakes, raising concerns regarding the prolonged stability of the wreck. An extensive series of field measurements were made during the summer of 2015 with the aim of differentiating between natural hydrological processes present at this site and human-derived water movements during the summer visitor season. There is a high-degree of natural current variability from processes as diverse as wind-induced surface gravity waves, internal gravity waves, and diurnal flows due to differential heating. Our results show that summer circulation driven by internal gravity waves derived from upwelling, surface waves, and differential heating was insignificant with respect to sediment resuspension and thus unlikely to produce the observed scour around the shipwreck. Scour is most likely caused by energetic winter storms, which should be a focus of future studies. While vessel induced currents were detectable at the shipwreck, they were no larger than the normal summer hydrodynamic variability, thus suggesting that management efforts continue to protect the site generally.

## Introduction

Ships have long plied and risked the world's waters, with over 3 million voyages ending in wreck [1]. Although lost from service, many shipwrecks continue to be recognized and valued for their cultural and historical significance, providing a tangible connection to the maritime cultural heritage of an area. In the Laurentian Great Lakes, there are over 6,000 shipwrecks [2], with about 1,000 within ready access of divers and boaters (e.g., [3]) and several hundred conserved and presented within protected areas such as the Thunder Bay National Marine

**Data Availability Statement:** The data underlying the results presented in the study are available on Zenodo (DOI: 10.5281/zenodo.5129267).

**Funding:** This work was supported by grants from the Great Lakes Protection Initiative (GLPI) (MW), the NSERC Discovery program under grant RGPIN-2016-06542 (MW), and Parks Canada (SP). The funders had no role in study design, data collection and analysis, decision to publish, or preparation of the manuscript.

**Competing interests:** The authors have declared that no competing interests exist.

Sanctuary in Lake Huron, USA and Wisconsin Shipwreck Coast National Marine Sanctuary in Lake Michigan, USA (e.g., [4, 5]). Indeed, a new Marine Sanctuary is also being discussed for eastern Lake Ontario, with specific aim "to manage a nationally significant collection of shipwrecks and other underwater cultural resources" [6]. While conserving a shipwreck in-situ is the preferred management approach [7], such a context continues to expose the resource to environmental factors that can contribute to its deterioration [8–10]. Preservation and maintenance of the structural integrity of submerged cultural resources is affected by a variety of hydro- physical, chemical, and biological factors. Physical factors include waves, currents, temperature, and depth, as well as human impacts [11]. Chemical factors include salinity, pH, and dissolved oxygen levels [11]. Biological factors include bacteria, fungi and various other organisms including Dreissenid mussels [11, 12]. All these factors interact in complex and non-linear ways, and can challenge the effectiveness of conservation efforts, which can be particularly concerning within those areas established and managed to protect and celebrate such submerged cultural resources. One challenge for conservation authorities is how to manage access to such shipwreck sites–in particular making sure that there are no strong water currents from vessels that could damage these underwater structures. With the increasing interest in such shipwrecks in the Great Lakes it is important to understand how currents induced by vessels compare with the intrinsic variability of natural currents.

Fathom Five National Marine Park (FFNMP), Lake Huron, Canada is one such protected area facing the challenge of managing visitor access with the preservation of culturally important shipwrecks (Fig 1A). Fathom Five Provincial Park was established in 1971 and slowly transformed the small community of Tobermory (Fig 1B) from a fishing village into one of Canada's premier recreational diving destinations, as well as a tourist destination due to the glass bottom tour boats during the summer visitor season (i.e., June–September) [13]. The provincial park was later transferred, along with the local islands of Georgian Bay Islands National Park to FFNMP. In 1987 the FFNMP establishment agreement was signed, and Parks Canada became the steward of its first site to be managed under the National Marine Conservation Area program [14, 15]. From the earliest days through today, a long-standing cultural resource management priority for FFNMP has been the conservation and preservation of the Sweepstakes and the maritime heritage it represents (Fig 1C–1F). The hull of the wooden sailing vessel has rested upright and nearly intact within a few meters of the water surface since 1885 and is perhaps Ontario's most photographed and popular shipwreck, with over 100,000 tour boat visitors and divers every summer (Parks Canada, unpublished data). With the passage of time, this iconic shipwreck has required various management interventions, including physical stabilization, monitoring, and restrictions on vessel activity, to maintain it in a safe and desirable state (e.g., [16, 17]). Vessel activity is restricted both in terms of timing and duration that tour boats can operate over the Sweepstakes, as well as enforcing a no-wake zone to minimize boat-induced currents. A major concern for stability of the Sweepstakes is lakebed scouring, which is particularly noticeable on the portside (see the faint ring, Fig 1C). Therefore, it is an important management goal to differentiate the natural and anthropogenic causes of sediment scouring due to water currents.

A preliminary field observation in a technical report by Boyce [18] suggested that sustained aggressive tour boat operation had the potential to lead to scouring, based on a short record (approximately 2 h, comprised of 6 tour boat passes over the shipwreck) with a single vector-averaging acoustic current meter at the portside of the Sweepstakes (in a similar location as our HR-ADCP) in 1994. However, they did not determine the relative strength of wind-driven currents, gravity flows due to upwelling events, surface wave orbital velocities, and flows induced by the wakes of tour boats under normal operating conditions. Furthermore, Boyce [18] observed isolated bottom current velocity peaks of 11.2 cm s$^{-1}$ and 17 cm s$^{-1}$ in 1993 and

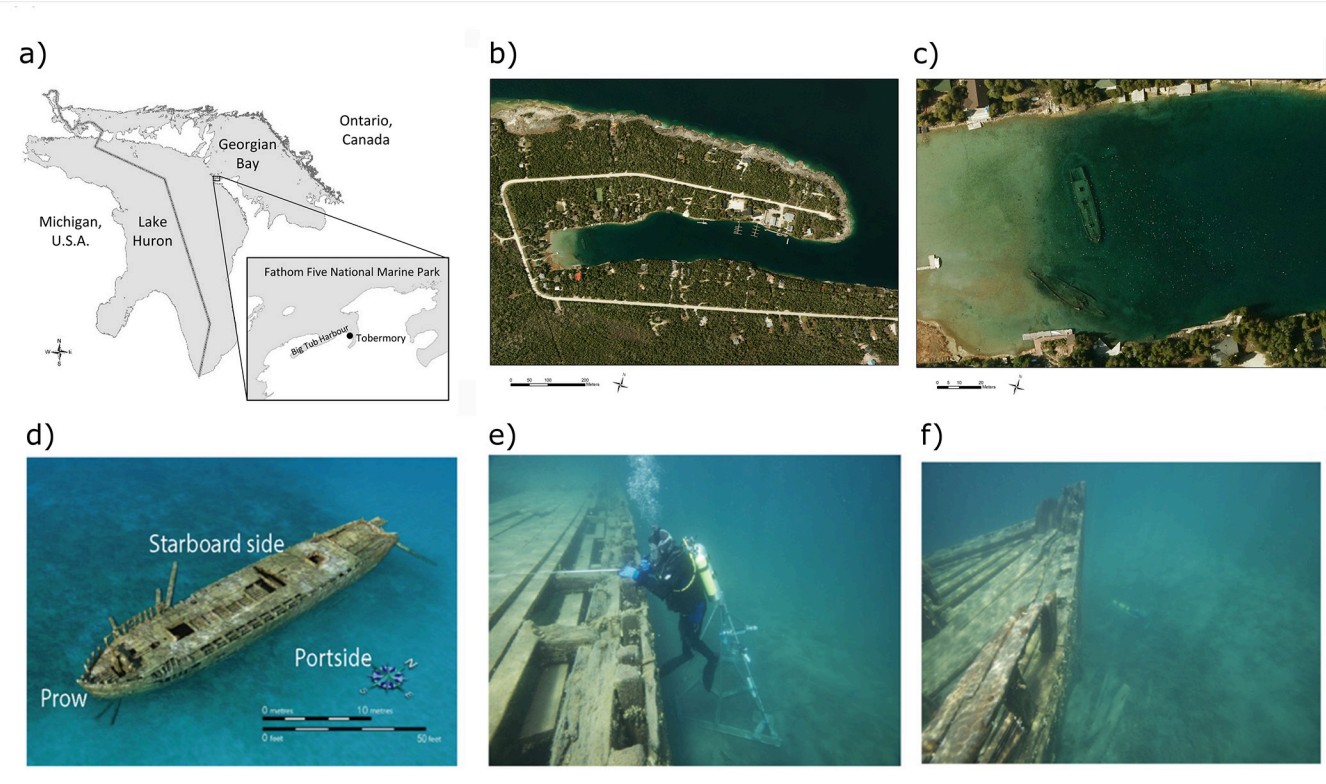

**Fig 1. The geographical location and the views of the Sweepstakes in Big Tub Harbour of Fathom Five National Marine Park near Tobermory, Ontario.** (a) A map of Lake Huron and Georgian Bay, Fathom Five National Marine Park, and Big Tub Harbour. (b) Aerial imagery of Big Tub Harbour. (c) Close up of western end of Big Tub Harbour, showing the two shipwrecks, namely, the Sweepstakes (1867–1885) and located to the south, the City of Grand Rapids (1879–1907). A ring of erosion is visible around the Sweepstake where no aquatic vegetation is growing and sand is exposed. While the Sweepstakes is a fully intact underwater shipwreck, the City of Grand Rapids has only the remains of its broken hull and machinery. (d) A sketch of the Sweepstakes. The prow faces south while the portside faces east towards Georgian Bay. (e) An underwater photograph of portside of the Sweepstakes, with the tripod visible behind diver. (f) An underwater photograph of the prow of the Sweepstakes, the yellow acoustic Doppler profiler (ADP) is visible in background on the bed. Reprinted from SWOOP 2010 imagery under a CC BY license with permission from Parks Canada, original copyright 2010, the Queen's Printer for Ontario.

1994 respectively, comparable to the 12 cm s$^{-1}$ minimum velocity that they calculated was required to initiate sediment movement. This isolated study provided the basis to support Parks Canada's management policies at the time. Several decades later the management context has changed, as there has been a notable increase in vessel size (~40% larger) and frequency of use.

Although the wreck will eventually collapse [17], the need to differentiate natural versus human-derived water movements during the summer will influence best management practices to conserve and manage the site today. While there has been considerable work done on the effects of scour around shipwrecks in energetic tidal systems in marine settings [19], we are not aware of analogous work done in lower-energy freshwater lake systems.

Natural water movements around the Sweepstakes in Big Tub Harbour can be attributed to wind-induced surface gravity waves, internal gravity waves, and gravity flows generated by differential heating. Previous studies of the thermal variability in FFNMP have shown that large-scale internal waves on the summer thermocline are ubiquitous and can have greatest temperature variability at depths of 10–20 m with periods of oscillation between 12 to 24 h [20]. Differential heating in an aquatic system with a sloping bottom can also create horizontal temperature (i.e., density) gradients that drive dense gravity currents flowing downslope [21]. There are numerous deep embayments in the Great Lakes with similar dynamics to this study.

For instance, the water flows in the Toronto Harbour (in Lake Ontario) are analogous to the flows we measured in big Tub Harbour, in that both are protected embayments connected to great lakes. In particular, Toronto harbours experiences periodic water currents up to 10 cm/s driven by 1-hour surface seiches, as well as periodic intrusions of cold stratified water [22, 23]. Similar flows are also seen in the many hundreds of embayments that exist behind the coastal archipelago along the eastern shore of Georgian Bay [24].

The major source of human derived water movements is thought to be the propeller and jet drive wash from boats that could drive increased water currents and pressure perturbations in the water column. While the previous study [18] suggested propeller wash could play a significant role under specific circumstances, it is unclear under the current management system what effect tour boats have on resuspension of bottom sediment. Additionally, recreational divers swimming near the shipwreck are a possible source of human-derived water movements and erosion.

In this manuscript, we aim to determine the relative magnitudes of natural and human-derived water movements during the peak visitor season that could influence the structural integrity of the Sweepstakes. We use detailed field measurements of water currents at three locations, pressure and temperatures in the vicinity of the Sweepstakes, and video observations for biological activities and tour boat activities acquired in the summer of 2015. Specifically, we quantify and differentiate natural versus human-derived water movements at the Sweepstakes and determine if there is any significant increase in water currents during the peak summer tourist season. We further assess the observed currents for their potential to initiate sediment movement and drive scour and erosion around the shipwreck.

The article is structured as follows: the study site, field measurements, data processing and analysis techniques are presented in the Methods section, followed by a summary of the observations and analysis in the Data and Results section. A discussion of the implications and limitations of our findings are presented in the Discussion section. A brief summary of the study and our findings are provided in the Conclusion section.

## Materials and methods

### Study site

Big Tub Harbour (81°40'38.67"W, 45°15'22.21"N) is located within Fathom National Marine Park on Georgian Bay (area of 15,000 km²), Lake Huron (area of 44,000 km²) [25]. It is a sheltered harbour with a rectangular shape, approximately 700 m long and 100 m wide with a mean depth of 12 m (Figs 2 and 3). To the east is the small town of Tobermory, located around the commercial port of Little Tub Harbour. The lakebed of Big Tub Harbour is composed of spatially discrete patches of silt, silty-sand, and sand and the harbour walls are dolomite bedrock. At the head of Big Tub Harbour rest two shipwrecks, the Sweepstakes and the City of Grand Rapids. The Sweepstakes (1867–1885) was a 36 m long two-masted wooden schooner, which, on the evening of 23 August 1885 struck a rock off Cove Island (located 3 km to the north) and sank stern first in shallow water. Weeks later it was salvaged and towed to Big Tub Harbour and eventually laid up and abandoned in approximately 7 m of water where her nearly intact hull remains today [26]. Also lying in the sand at the head of the harbour just south of the Sweepstakes is the broken, fire–gutted remains of the steamer City of Grand Rapids (1879–1907).

**Big Tub Harbour bed sediment and vegetation structure.** The lakebed composition was mapped by classifying data from a 2007 multi-beam backscatter survey of the harbour (Fig 2). The analysis was trained using Ponar grab and video samples of the harbour bed. The classification provides a coarse sediment structure and general distribution of submerged aquatic

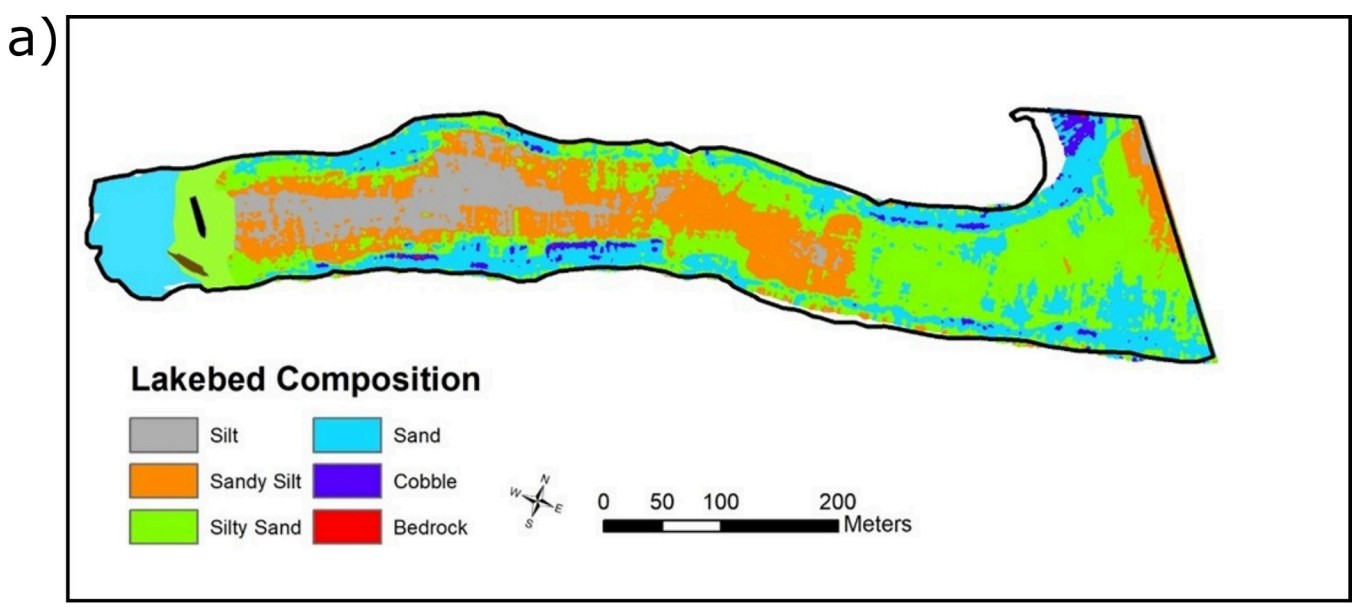

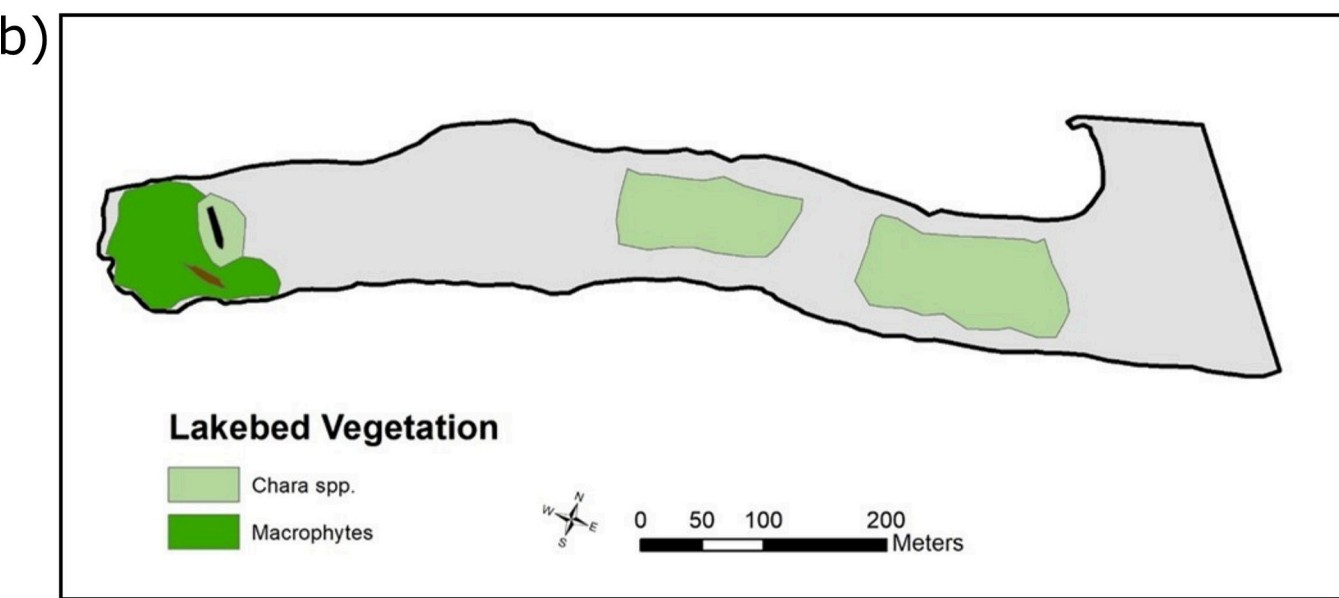

**Fig 2. Composition of harbour bed sediments and aquatic vegetation in Big Tub Harbour.** (a) Sediments and (b) aquatic vegetation. The location of the two shipwrecks are shown as silhouettes. The classification provides a coarse sediment structure on the harbour bed and general distribution of submerged aquatic vegetation. Around the two shipwrecks at the head of Big Tub Harbour, the bed is dominated by silty sand and *Chara sp*. vegetation.

vegetation. Silty sand dominates near the shipwreck site, which Boyce [18] found to be in the range of 125–200 microns. The major forms of benthic vegetation are *Chara spp.* and macrophytes (e.g., *Myriophyllum spp.*, *Potamogeton spp.*).

### Field measurements

The field data collection campaign at Big Tub Harbour ran from 5 May 2015 to 13 October 2015, and was jointly undertaken by Parks Canada, Environment Canada, and the University

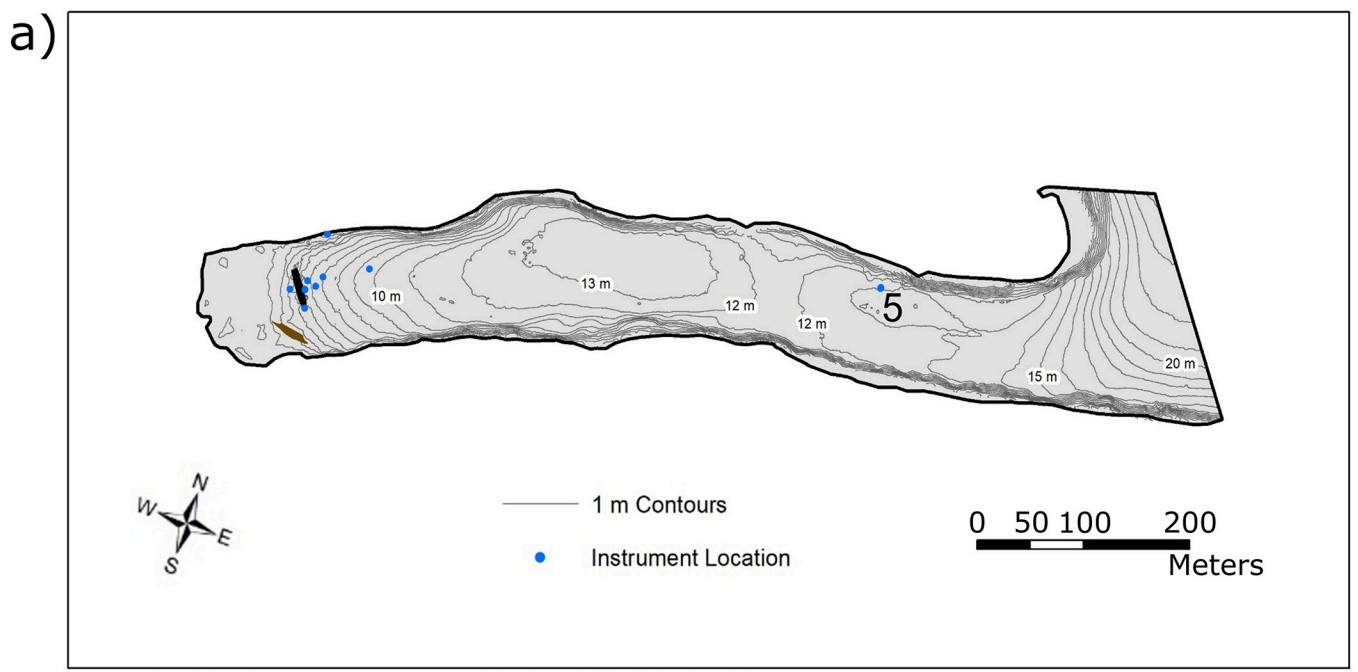

**Fig 3. Bathymetry of Big Tub Harbour and geographical locations of the field instruments relative to the Sweepstakes.** The Sweepstakes wreck (black silhouette) is visible with the prow pointing to the southeast. The numbers correspond to individual instruments as follows: 1: Float 1 (one thermistor), 2: Float 3 (two thermistor loggers), 3: Float 4 (two thermistor loggers), 4: Float 5 (two thermistor loggers), 5: Marine Operations Base (MOB) chain (13 thermistor loggers), 6: HR-ADCP 7: AWAC (also a pressure sensor) 8: ADP and downward looking camera), 9: On shore surface camera. Bathymetry contours in a) and b) are 1 m.

of Toronto. The purpose of the monitoring was to study summer water movements and differentiate natural movements (e.g., gravity currents, waves, and seiches) from the motions forced by the vessels around the Sweepstakes. The locations of the instruments relative to the Sweepstakes wreck are given in Fig 3 and a summary of instruments used are presented in Table 1. Videos of the instrument setups and locations are available in the Supporting Information.

To measure the water column temperature, HOBO Tidbit v2, UTBI-001 thermistors were used. Three arrays (floats 1, 3, and 4) were deployed around the Sweepstakes (locations given as 1, 2 and 3 in Fig 3) and one array (float 5) was deployed a few meters east from the Sweepstakes (location 4 in Fig 3). Three arrays (floats 3, 4, and 5) contained two thermistors installed 0.5 m and 1.0 m above the harbour bed. The remaining array (float 1) only contained a single thermistor installed 0.5 m above the harbour bed. The loggers were deployed at approximately 08:00 EST, 12 June 2015 and retrieved at approximately 12:00 EST 26 August 2015. Another large array (Marine Operations Base-MOB chain) of thirteen HOBO Tidbit v2, UTBI-001 thermistors was placed near the mouth of the harbour (given as location 5 in Fig 3) to record the temporal fluctuations of the harbour's water column temperature. The array was deployed from 5 May 2015–22 May 2015. The data record started again at approximately 12:00 EST, 23 May 2015 and was retrieved at approximately 10:30 EST 17 September 2015. The thermistors recorded the water temperature every 5 minutes with a resolution of 0.02°C and an accuracy of ± 0.21°C.

One acoustic Doppler profiler (SonTek ADP, S/N M945) was installed in 5.5 m water depth approximately 5 m off the prow of the Sweepstakes, pointing upwards (location 8 in Fig 3, also visible in Fig 1F). The ADP, with 1500 kHz frequency, was programmed to ping as rapidly as possible and record the 30-seconds average current velocity in all three directions in every 2-minutes. This allows the ADP to gain 30-seconds of measuring followed by 90-seconds of inactivity in each 120-seconds. The ADP had a blanking distance of 0.4 m. The ADP recorded the average current readings for seven 0.5 m bins, from 0.8 m to 4.3 m above harbour bed. The

**Table 1. Location, measured water quality parameter, and observation frequency of instruments deployed in Big Tub Harbour, 2015.**

| Position on Map | Instrument Name | Measured property | Sampling interval /Frequency | Depth and other information |
|---|---|---|---|---|
| 1 | Float 1 | Temperature | 5 mins | One logger at 0.5 m above the harbour bed. |
| 2 | Float 3 | Temperature | 5 mins | Two loggers at 0.5, 1 m above the harbour bed. |
| 3 | Float 4 | Temperature | 5 mins | Two loggers at 0.5, 1 m above the harbour bed. |
| 4 | Float 5 | Temperature | 5 mins | Two loggers at 0.5, 1 m above the harbour bed. |
| 5 | MOB chain | Temperature | 5 mins | 13 loggers at 0.5, 1, 2, 3, 4, 5, 6, 7, 8, 9, 10, 11, 12 m above the harbour bed. |
| 6 | HR-ADCP | Currents, pressure | 1024 sample per 5 min burst interval | Down looking and 1.5 m above the harbour bed. |
| 7 | AWAC | Currents | 1024 samples per 20 min burst interval | Upward looking and 6.5 m of from the surface water. |
| 8 | ADP | Currents, camera for biological activities | One sample per 2 min burst interval | ADP is Upward looking and 5.5 m of from the surface. The camera is downwards angle away from the wreck and installed 6.5 m of from the surface. |
| 9 | Surface Camera | Tour boat visitation times | continuous | On shore |

ADP was deployed on 24 June 2015 at around 10:30 EST and retrieved on 21 August 2015 at approximately 16:30 EST. An acoustic wave and current profiler (Nortek AWAC, 600 kHz) with acoustic surface tracking (AST) was installed approximately 5 m off the port side (east) of Sweepstakes (location 7 in Fig 3). It was placed in an upward facing configuration in approximately 6.5 m of water. In 20-minute intervals, it sampled at 1 Hz for 17.06-minutes (i.e., 1024 samples per burst). The AWAC started recording on 23 June 2015 at 16:00 EST and ran until 8 July 2015 at 23:00 EST. The AWAC has a blanking distance of 0.5 m and current velocities were measured in 0.5 m bins. The AWAC has an accuracy of 1% of the measured value ± 0.5 cm/s. The AST feature allows for accurate measurements, 0.1% of full scale of the water surface elevation, to measure surface waves or wakes. The third acoustic profiler, a high-resolution acoustic Doppler current profiler (HR-ADCP, Nortek Aquadopp HR) was located close to the port side of the shipwreck (location 6 in Fig 3). It was mounted in a downward-looking configuration on a tripod 1.5 m above the harbour bed, visible in Fig 1E. The ADCP has a blanking distance of 0.10 m. Current velocities were measured in 48 bins, each 0.03 m, spanning from the harbour bed to 1.4 above the harbour bed. With a sampling frequency of 4 Hz, the instrument samples 1024 times (in 256 s) in each 300-second burst interval. The instrument was deployed on 23 June 2015 at 12:00 EST and the last measurement was recorded on 13 October 2015 at 11:03 EST.

An underwater video camera was installed on the same tripod as the HR-ADCP, on the port side (east) of the Sweepstakes in 6.5 m of water (location 8 in Fig 3). The camera was oriented to look at a downwards angle away from the wreck. Video footage was recorded, with a few gaps, on 29 June 2015 and then fairly continuously from 3 July 2015 until 14 July 2015. Due to poor visibility at night, it was only possible to analyze video taken during the day resulting in a total of approximately 100 usable hours of underwater footage. A second camera (Plotwatcher Pro, Model TLC-200-C) was placed on shore from 25 June 2015–3 July 2015 looking over the Sweepstakes wreck site, to make a record of exactly when tour and other boats were present above the shipwrecks (location 9 in Fig 3). The underwater video camera footage allowed us to capture any sediment re-suspension events and corresponding possible causes that happened in the water column. For instance, we captured biological activities such as round gobies (*Neogobius melanostomus*) digging in the sediments causing localized re-suspension that would not be observable in the water temperature and current records.

A pressure logger was mounted on HR-ADCP frame 0.75 m above the harbour bed (45˚ 15.316'N, 081˚ 40.849'W, location 6 in Fig 3). It continuously sampled at 2 Hz from 23 June 2015 at 16:00 EST and ran until 12 September 2015 at 02:50 EST. Hourly mean wind speeds and direction were obtained from Environment Canada's Tobermory Airport Weather Station (Tobermory RCS, WMO ID 71767). The station is located at 45˚14'00.000" N and 81˚ 38'00.000" W.

## Data processing

To evaluate the major sources of energy–natural or human-derived water movements–that could be responsible for scouring around the Sweepstakes, we used time series plots of temperature and bottom currents. We estimated how much current variability is due to natural physical processes such as wind-induced surface gravity waves, internal gravity waves, and diurnal flows due to differential heating. We extracted water currents driven by human interaction as a function of prop-wash induced currents. We then compared the bottom currents with respect to natural variability and human interactions to identify the major sources of energy that could potentially be responsible for scouring.

Magnitude of near-bed currents that could potentially result in scour and erosion were quantified using measurements acquired at the prow and port side of the Sweepstakes. Velocity data processing was divided into a few steps. First, we plotted the time series of the east-west and north-south velocities acquired at all acoustic current meters located in the vicinity of the shipwreck to visually identify the bottom currents. The HR-ADCP records data up to 1.4 m from the harbour bed. Hence, for comparison purposes we only consider the currents' variability in the depths up to 1.5 m from the harbour bed. If the bottom currents show a barotropic variability, such that there is no vertical velocity gradient, we can average the velocity bins up to 1.5 m from the harbour bed. Then, we used Fast Fourier Transform (FFT) analysis to identify the dominant periods of the bottom currents in the vicinity of the shipwreck. The dominant periods reveal relevant peaks of natural forcings. In this analysis, we de-trended the speed data, and then used the Welch [27] algorithm where the power spectrum is estimated by dividing stationary data into segments. The number of segments depend on the length of the time series. Thus, we found the modified periodogram for each segment that expresses the uncorrelated estimates of the spectra. To obtain the average of the modified periodograms, the segments are multiplied by a window function (Hanning window) with a 50% overlapping technique to reduce the variance of the periodogram. To evaluate the currents induced by the propeller wash from the tour boats, we divided data in to two windows; times that the boats were present and the times when the tour boats were absent. We then plotted the histogram of the current speeds with respect to the time windows selected. To compare the results, we used probability of speed occurrences which varies between 0 and 1. We hypothesize that if the bottom currents show an increased variability when the boats were present, then the propeller wash induced currents could contribute to scouring of the sediment and thus might compromise the structural integrity of the Sweepstakes.

Cold water intrusions could be a source of energy for scouring. These could be caused by upwelling of cold water from Lake Huron into Big Tub Harbour. To determine if such intrusive currents are present, we monitored water temperatures at mouth of harbour and near the Sweepstakes. The cold intrusive events are identified as a drop in water temperature by 5–8°C in the space of few hours. Often, these cold-water upwelling events are driven by strong local winds or by internal gravity waves induced by distant wind events [20]. Thus, we first plotted the time series of the temperature measurements to visualize the spatial and temporal variability of the temperatures at different depths. We then adapted the Continuous Wavelet Transform (CWT) method [28] to determine the times that upwelling is significant. CWT expands the time series to time-frequency domain. Next, we used the same spectral analysis described above to determine the dominant periods related to temperature variability. To evaluate the importance of episodic upwelling events on bottom currents driven by internal gravity waves that can scour the bottom sediments around the shipwreck, the time window is split into times when upwelling occurs and when it does not. We then compared the histograms of horizontal bottom currents at the times corresponding to upwelling and non-upwelling. For the comparison purpose, the frequency in the histogram was normalized. If the bottom currents show a significant increase in variability during the identified upwelling events, one could assume that the circulation is driven by the gravity flows induced by upwelling. In addition, there could be standing surface waves, or seiches, in the harbour, similar to those seen at nearby sites in FFNMP that were visually observed to lead to significant water currents [29].

To account for the discussion of propeller- and jet drive-wash induced forcing, we applied the spectral analysis obtained from the FFT (described above) to the pressure measurements acquired from the sensor that was attached to the HR-ADCP. The FFT results were used to examine the dominant frequency of any seiche-induced oscillations.

## Estimation of seiche periods in harbour

Big Tub Harbour is a shallow, open-mouth, long, and narrow basin with a rectangular shape that potentially could support standing wave oscillations. The frequency of these waves can be made by assuming the depth of the harbour is approximately a constant and there are vertical walls on the sides. Thus, periods of the eigen (natural) modes of the standing oscillations in such an open basin can be described using the classic Merian formula [30]

$$T_n = \frac{4L}{(2n + 1)\sqrt{gH}} \tag{1}$$

where, $T$ is the period, $n$ is the modes of the oscillations, $g$ is the gravitational acceleration (~ 9.8 m/s$^2$), and $H$ is the water depth. The first mode ($n = 0$) is known as the Helmholtz resonance mode such that, Eq (1) becomes

$$T_0 = \frac{4L}{\sqrt{gH}} \tag{2}$$

For instance, Big Tub Harbour is approximately 690 m long ($L$) and 12 m deep ($H$). Thus, the Helmholtz resonance period ($T_0$) is computed as 4.2-minutes (Eq 2).

## Calculating high frequency pressure perturbations

To determine the pressure perturbations generated by the high frequency waves near the Sweepstakes, the measurements have a high-pass filter applied at 4-minutes. The high-pass filter at 4-minutes removes any variability caused by natural modes of oscillations in the harbour and retains only high frequency events (those with periods shorter than 4 minutes). The high-pass filtered amplitude of the pressure perturbation variability caused by the water level fluctuations (such as from high frequency waves) will then be compared with the times that the boats were present and absent. The pressure perturbation is defined as

$$P' = P_{total} - P_{hydrostatic}, \tag{3}$$

where, $P'$ is the pressure perturbation, $P_{total}$ is the high-pass filtered total pressure measured by the pressure sensor attached to the HR-ADCP located at the port side of the Sweepstakes, $P_{hydrostatic}$ is the hydrostatic pressure (= $\rho gH$), $\rho$ is the water density (~1000 kg/m$^3$), $g$ is the acceleration due to gravity (~ 9.8 m/s$^2$), and $H$ is the total water column depth. As $\rho$ and $g$ are constant over short periods, the pressure perturbation $P'$ is usually reported as an equivalent depth of water in metres. The time series of high-pass filtered pressure perturbation amplitude is then compared with the time series of wind speeds with direction as a proxy for when surface waves would likely have been large.

## Critical velocity for sediment transport

Water flowing over the habour bed exerts a stress on the bottom sediments which, when strong enough, results in sediment transport, which is often separated into suspended load and bedload modes [31]. While bedload transport occurs in a series of short hops, suspended load transport typically has much longer trajectories due to the particles suspended in the water column being advected by the water currents [31]. Based on work in the New York Bight, [31] found that very fine sand particles (0.063 mm in diameter and smaller) are most often transported in the suspended load mode, while sand particles with a diameter of 0.125 mm or larger typically move as bedload, as the stress required to keep them in suspension is greater that the stress required to initiate movement. With particle size of the fine-grained silty

sand in the vicinity of the Sweepstakes in the range of 31–250 microns (0.03–0.25 mm) [18], sediment could potentially be transported by either suspended load or bedload modes.

Current velocity required to initiate transport can be estimated as $u(z) = \frac{u_*}{\kappa} ln\left(\frac{z}{z_o}\right)$ where $u$ (z) is the current velocity at height $z$ above the bed, $u_*$ is the shear velocity at the bed, $\kappa = 0.4$ is the von Karman constant, and $z_o \approx d/30$ is the roughness length scale (where $d$ is the representative particle diameter) [31]. This represents a rough estimate, as the relationship between shear velocity and current velocity is complex and depends upon many factors including grain size, bottom roughness, and flow type (i.e., steady, oscillatory, or a combination of both), while the critical shear stress required to initiate movement is itself influenced by various abiotic and biotic factors [31]. Nevertheless, using the above equation suggests a current velocity on the order of 27 cm s$^{-1}$ at 1.5 m above the harbour bed is required to initiate suspended load transport of the very fine-grained sediment in the vicinity of the Sweepstakes (Based on experimental work by Butman [32], sand with a particle diameter of $d = 63$ microns has a critical shear suspension velocity of $u_* = 0.8$ cm s$^{-1}$) [31]. This corresponds to a current velocity of 12 cm s$^{-1}$ at a height of 1 mm from the bed, congruent Boyce's [18] bottom orbital velocity estimate of 12 cm s$^{-1}$ required to initiate sediment movement.

## Data and results

The goal of the study was to differentiate between the natural hydrological processes and human-derived water movements from boat traffic present in the vicinity of the Sweepstakes shipwreck site. The analysis identifies natural current variability from processes such as wind-induced surface gravity waves, internal gravity waves, and diurnal flows due to differential heating. Due to the detectable variability caused by tour boats, we compared the effect of propeller-wash induced bottom currents with respect to that caused by natural variability.

### Thermal structure

During summer the waters around the Sweepstakes are thermally stratified (Fig 4). Water temperature time series extracted at Floats 1, 3, 4, 5 located near the Sweepstakes and MOB chains located near the Parks Canada dock (toward the mouth of the harbour) show the spatial and temporal evolution of the thermal structure in Big Tub Harbour (Fig 4). During the deployment period the surface waters in Big Tub Harbour (Fig 4A) gradually warm from 10˚C, reaching a maximum temperature of 20˚C in early-to-mid August (around DOY 221–230). A similar warming was observed in the bottom temperatures in the direct vicinity of the Sweepstakes (Fig 4B–4E). For instance, on DOY 180 (29 June 2015), the mean daily temperature 1 m from the harbour bed at the MOB chain was 7.6˚C and rose to 17.6˚C by DOY 229 (17 August 2015). This corresponds to an average warming trend of ~0.2˚C per day (Fig 4A). Big Tub Harbour (with a mean depth of 12 m) is shallower than the depths where temperature variability is greatest in FFNMP (i.e., the depth at which the summer thermocline lies), which Wells and Parker [20] observed at 20 m depth.

Stratified water bodies often have strong currents associated with internal waves that can be a strong source of oscillating currents [33, 34]. To determine if such internal waves are important in driving upwelling events, the dominant frequencies are determined from a power spectrum analysis. The power spectrum of the temperature measurements at 0.5 m from the harbour bed acquired at temperature loggers in the vicinity of the shipwreck shows a strong semi-diurnal signal (Fig 5) which could be the semidiurnal lunar ($M_2$) tide, which has a period of 12.42 h. Another distinct mode was identified at 6.15 h which may be related to the cold-water intrusions at the bottom (upwelling). Further, the above calculated period of 6.15 h is

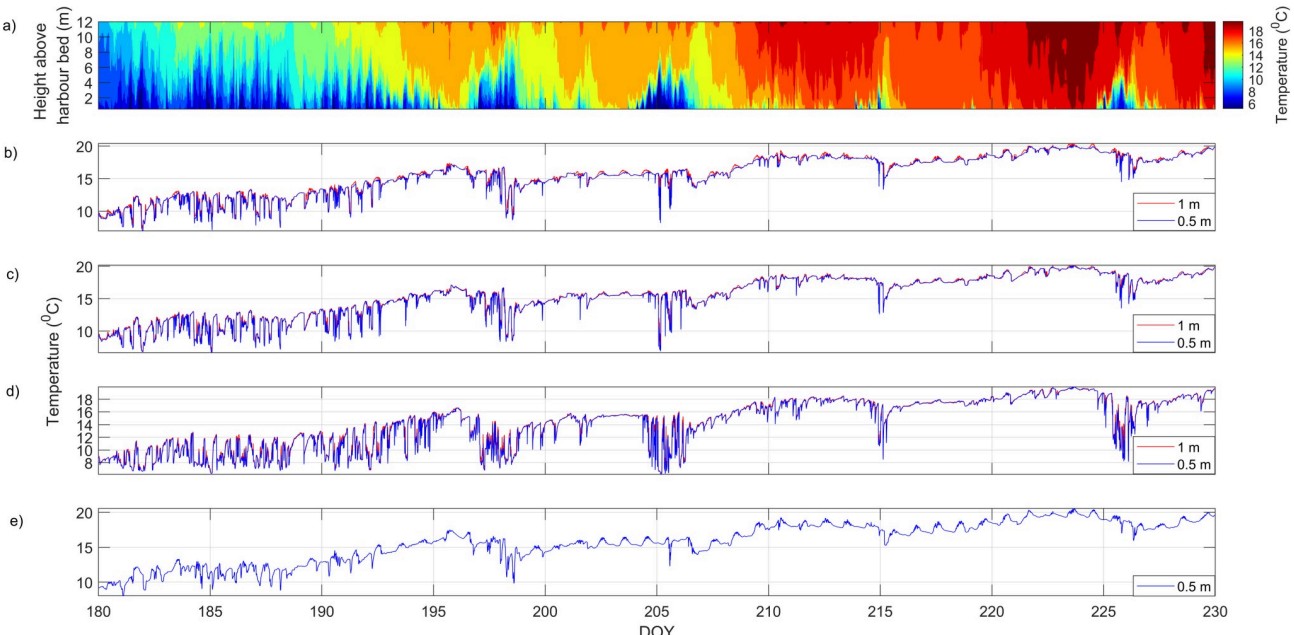

**Fig 4. Water temperature time series from 29 June (DOY 180) to 18 August (DOY 230), 2015.** (a) Contour plot of water temperature at MOB (location 5 in Fig 3). Note the strong upwelling signals in the deeper water, the strong daily warming signal near the surface, and the general warming trend as time progresses. (b) Temperature measurements at Float 3 (closest chain to the east of Sweepstakes). (c) Temperature measurements at Float 4. (d) Temperature measurements at Float 5. (e) Temperature measurements at Float 1. Note that the larger fluctuations in temperature at the lower thermistor (at 0.5 m from the harbour bed) are observed in all floats due to cold water upwelling.

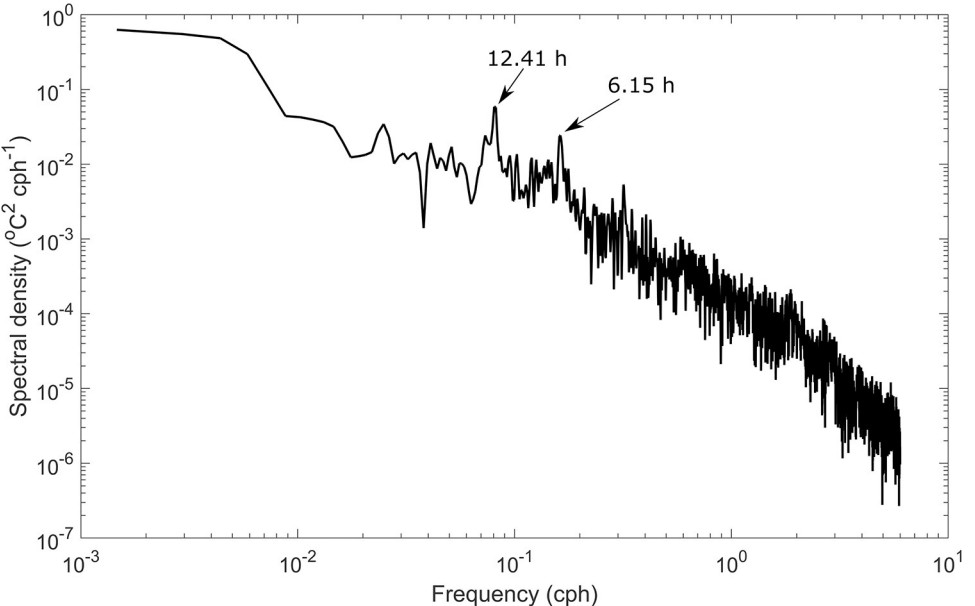

**Fig 5. Spectral density diagram for the de-trended bottom temperature (0.5 m from the harbour bed) acquired at float 3.** The record shows significant periods at 12.41 h (semidiurnal) and at 6.15 h. A similar behaviour is seen in all floats in the vicinity of the Sweepstakes shipwreck.

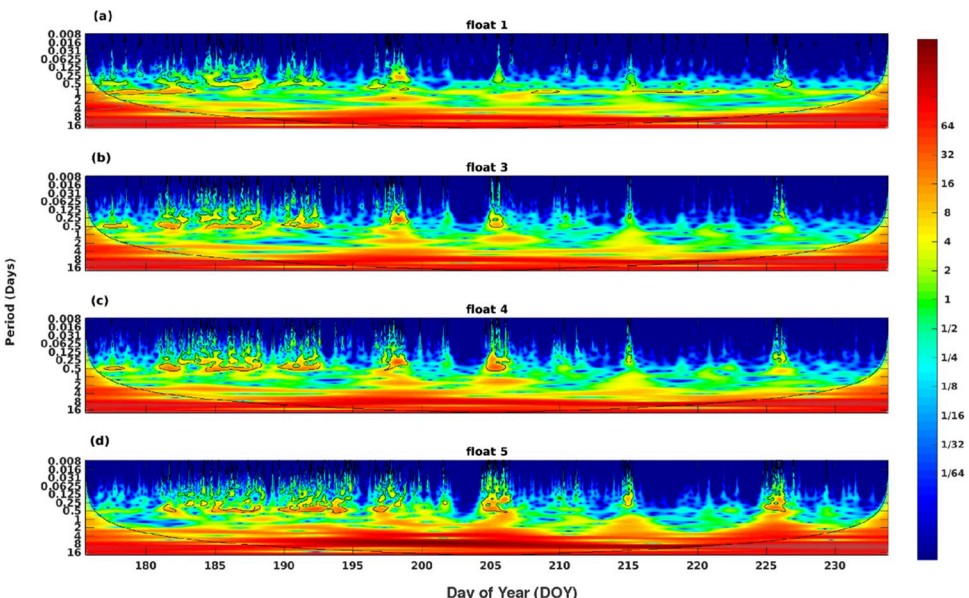

**Fig 6. Continuous Wavelet Transform (CWT) of the temperatures obtained at the bottom thermistors (0.5 m above the harbour bed).** (a) float 1, (b) float 3, (c) float 4, and (d) float 5. CWT expands the time series into time-frequency space. The record shows four distinct upwelling events on DOY 198, 205, 215, and 226 on all temperature records.

close to the H1 seiche mode (Lake Huron mode 1) documented in Schwab et al. [35] which is about 6.6 h. Thus, upwelling events in the temperature records may be driven by the free modes of oscillations attributed to Lake Huron seiches.

Upwelling of cold water from Lake Huron into the warmer waters of Big Tub harbour is observed frequently during the summer (Figs 4 and 6). Both the water temperature plots (Fig 4) and the continuous wavelet transform of the bottom temperature records (0.5 m above the harbour bed; Fig 6) show four distinct upwelling events on DOYs 198, 205, 215, and 226. During these coldwater intrusion events at the bottom, the water temperature quickly drops and rises again by 5–8°C over a few hours (Fig 4). Sharp decreases and subsequent increases in temperature observed at all temperature loggers (both at 0.5 m and 1 m from the harbour bed) near the Sweepstakes suggest upwelling events extend all the way to the head of the harbour and to at least 1 m above the harbour bed (Fig 4). These episodic upwelling events suggest that the waters of Big Tub Harbour are frequently exchanged with waters from Lake Huron.

## Effect of upwelling on water circulation

The frequent upwelling events have very striking drops in water temperature, but as they occur slowly they do not produce significant water currents. The variability of bottom currents during the four distinct upwelling events that were identified in the temperature time series (Figs 4 and 6) are compared with the variability during the times when there is no visible upwelling, which we refer to as non-upwelling events or periods (Fig 7). The comparison will quantify the strength of the flows driven by the internal gravity waves in the Big Tub Harbour. Bottom currents that show increased variability during upwelling events could suggest that gravity currents, driven by combination of differential heating and internal-seiches in Lake Huron, might potentially contribute to scouring around the shipwreck. However, if the maximum velocities or velocity variability does not change, then the effect of internal gravity waves

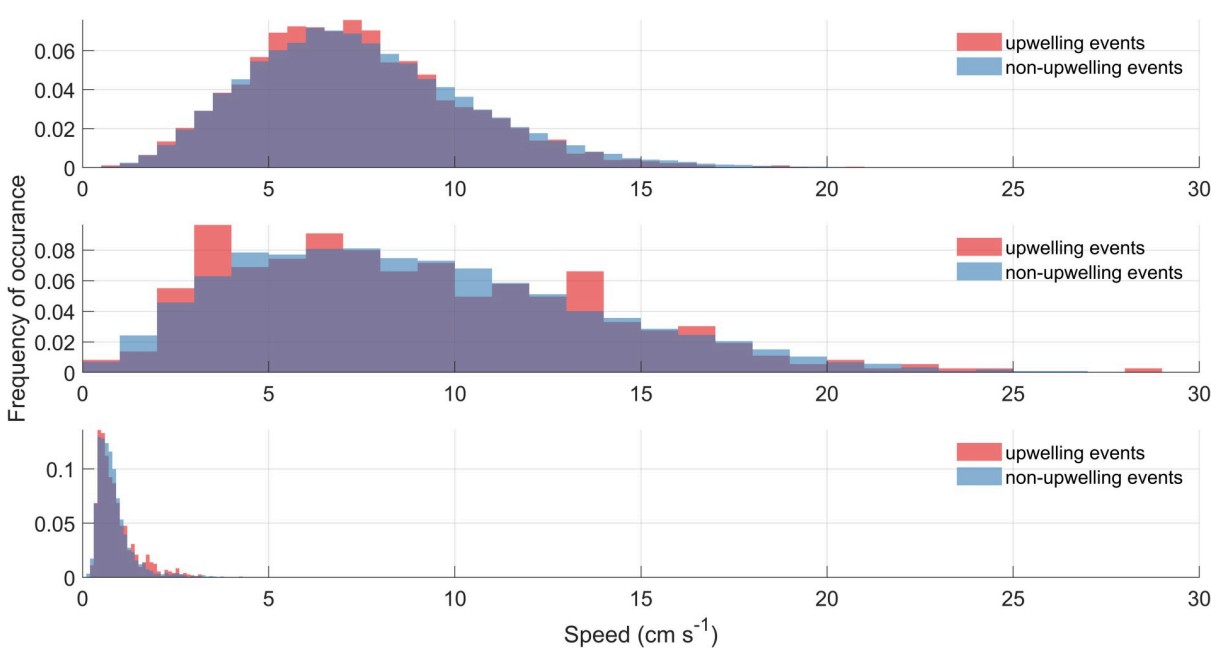

**Fig 7. Histogram analyses of bottom currents with the presence and absence of upwelling events.** (a) ADP located at the prow (b) AWAC located at the portside of the shipwreck, and (c) HR-ADCP located at the portside of the shipwreck. The y-axis value is the normalized frequency of speed occurrences between 0–1. Blue color denotes the bottom speeds during non-upwelling events and the red color represents the speeds of the bottom water currents during upwelling periods. The upwelling events are observed on DOY 198, 205, 215, and 226 (See Figs 4 and 6 for visualization of upwelling events).

on the flows is minimal. For comparison purposes, the distribution is normalized (Fig 7) to give the probability of speed occurrences. From visual analysis, the speed probability distributions do not indicate an increased variability during upwelling events (Fig 7). This is corroborated by the mean currents observed at the ADP (7.3 cm s$^{-1}$ and 7.5 cm s$^{-1}$ during upwelling and non-upwelling periods respectively), the HR-ADCP (0.9 cm s$^{-1}$ during both upwelling and non-upwelling periods) and the AWAC (9.0 cm s$^{-1}$ during both upwelling and non-upwelling periods) (Table 2). Resuspension is most likely to occur during high-velocity events, which we characterize as the top 10% of all observations (the 90$^{th}$ percentile). While the 90$^{th}$ percentile current speeds were typically faster during upwelling periods at the HR-ADCP (1.7 cm s$^{-1}$ compared to 1.4 cm s$^{-1}$), they were slower at the ADP (11.2 cm s$^{-1}$ compared to 11.6 cm s$^{-1}$) and the AWAC (15.6 cm s$^{-1}$ compared to 15.7 cm s$^{-1}$) (Table 2).

**Table 2. Bottom (1.5 m from bed) water current speed statistics during upwelling and non-upwelling periods in the vicinity of the Sweepstakes.**

| | ADP | | AWAC | | HR-ADCP | |
|---|---|---|---|---|---|---|
| | Upwelling | Non-upwelling | Upwelling | Non-upwelling | Upwelling | Non-upwelling |
| 5$^{th}$ percentile (cm s$^{-1}$) | 3.1 | 3.2 | 2.6 | 2.5 | 0.4 | 0.4 |
| 25$^{th}$ percentile (cm s$^{-1}$) | 5.3 | 5.4 | 5.2 | 5.4 | 0.5 | 0.5 |
| 75$^{th}$ percentile (cm s$^{-1}$) | 9.0 | 9.3 | 12.3 | 12.3 | 1.1 | 1.0 |
| 90$^{th}$ percentile (cm s$^{-1}$) | 11.2 | 11.6 | 15.6 | 15.7 | 1.7 | 1.4 |
| 95$^{th}$ percentile (cm s$^{-1}$) | 12.5 | 13.1 | 17.6 | 19.2 | 2.0 | 1.8 |
| Mean Speed (cm s$^{-1}$) | 7.3 | 7.5 | 9.0 | 9.0 | 0.9 | 0.9 |
| Std. dev. (cm s$^{-1}$) | 2.9 | 3.1 | 4.9 | 4.8 | 0.5 | 0.5 |

## Bottom currents in the vicinity of the shipwreck

Due to the orientation of the shipwreck in Big Tub harbour, different regions around the hull experience different water velocities, and hence different potential for erosion. To understand the spatial variability in water movements, we compared the current speed data at the prow, measured by the ADP (location 8 in Fig 3), and at the port side of the shipwreck, measured by two different ADCPs, namely, AWAC (location 7 in Fig 3) and HR-ADCP (location 6 in Fig 3). The HR-ADCP is the closest to the port side of the shipwreck while AWAC is few meters away from HR-ADCP, towards the harbour mouth (Fig 3). The velocity time series show an oscillatory motion at 1.5 m from the harbour bed (Fig 8). The oscillatory motion can be defined as a barotropic flow such that there is no vertical velocity gradient in the water column. The FFT analysis shows significant peaks at 23.75 h (~diurnal), and at 12.0 h (~semi-diurnal) for the mean bottom current speeds (i.e., 1.5 m from the harbour bed). Because of the barotropic motion, currents observed within 1.5 m of the harbor bed were averaged (Fig 9). Currents at 1.5 m from the harbour bed at the prow of the shipwreck, extracted from the ADP show a mean speed of ~7.5 cm s$^{-1}$, minimum of ~0.2 cm s$^{-1}$, a maximum of ~33 cm s$^{-1}$, and a range of ~33 cm s$^{-1}$ (Fig 9B). The speed calculated from velocity measurements acquired from AWAC shows a mean speed of ~9 cm s$^{-1}$ s (minimum, maximum, and the range are ~0.1 cm s$^{-1}$, ~31 cm s$^{-1}$, and ~31 cm/s, respectively; Fig 9C). The currents measured by the HR-ADCP near the port side of the Sweepstakes show a mean speed of less than 1 cm s$^{-1}$ but show some very brief periods (10 s) of higher speeds (Fig 9D). The maximum and the minimum speeds recorded by the HR-ADCP during the observation period are ~0.2 cm s$^{-1}$ and ~5 cm s$^{-1}$, respectively, with a range of ~4.5 cm s$^{-1}$. From these measurements, it is clear the current speed is an order of magnitude larger at the prow (Fig 9B) compared to the port side of the shipwreck (Fig 9C).

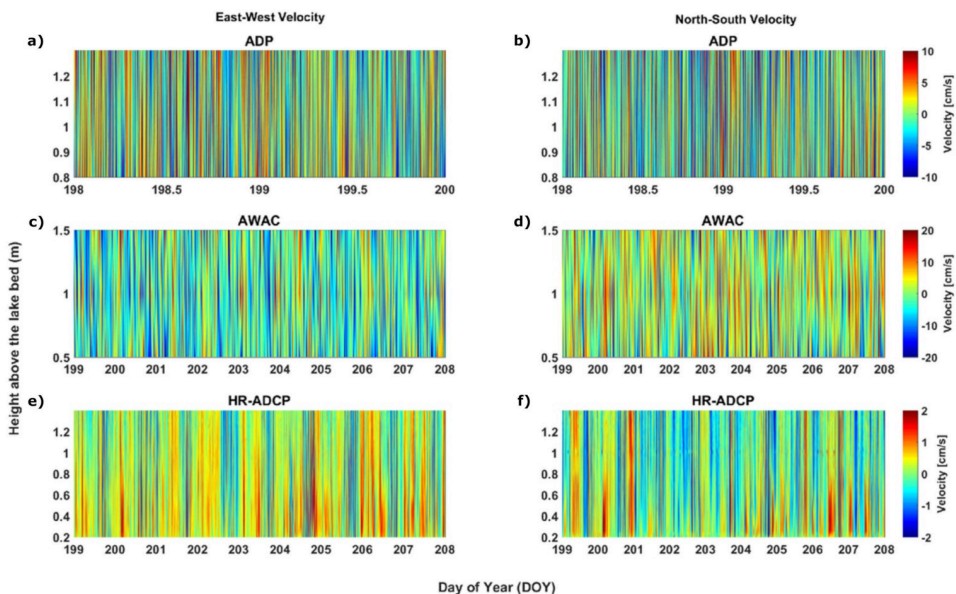

**Fig 8. The east-west and north-south velocities up to 1.5 m from the harbour bed.** (a) The east-west velocities extracted by ADP which is located at the prow, (b) The north-south velocities extracted by ADP (c) The east-west velocities extracted by AWAC on the side, and (d) The north-south velocities extracted by AWAC (e) The east-west velocities extracted by HR-ADCP located closest to the side of the shipwreck. (f) The north-south velocities extracted by HR-ADCP. The oscillatory motion in velocity distribution shows a barotropic flow (almost no vertical velocity gradients) in the bottom water column.

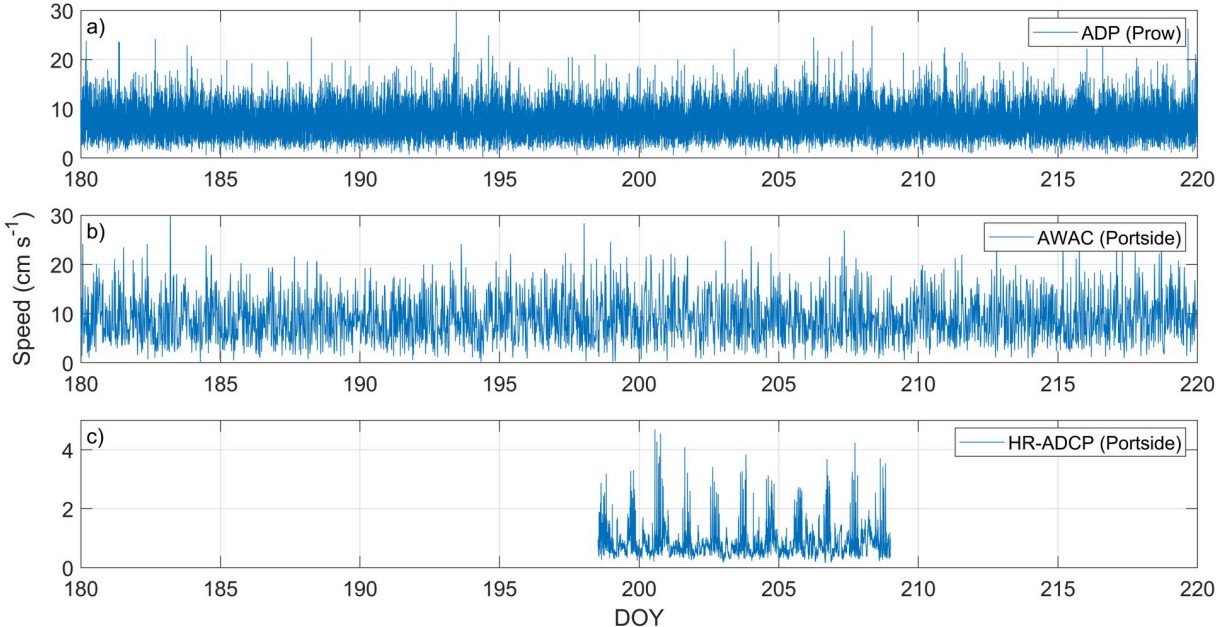

**Fig 9. The depth-averaged current speeds up to 1.5m from the harbour bed.** (a) ADP which is located at the prow, (b) AWAC on the side, and (c) HR-ADCP located closest to the side of the shipwreck (note the different scale on y-axis). The mean speed nearest to the shipwreck (HR-ADCP, subplot c) but located on the side was ~1 cm s$^{-1}$ while the measured mean speed approximately 5 m from the side (AWAC, subplot b) is ~9 cm s$^{-1}$. Measured mean speed at the prow was ~8 cm s$^{-1}$. The minimum speed required to initiate suspended load transport is approximately 27 cm s$^{-1}$.

## Effect of tour boat propeller wash on water currents

The most important result in this paper is to determine if currents induced by tourist boats produce currents that have similar magnitudes to natural currents present in the area, and if they are capable of causing erosion. The effect of tour boat propeller wash on water currents was evaluated by comparing the bottom water currents in the immediate vicinity of the Sweepstakes when boats were present and when they were not. We hypothesize that if the bottom water currents are significantly faster, or if there is significantly more turbulence in the water column when boats are present then there should be a correlation between boat activity and water velocity around the shipwreck. If, however, water currents are not significantly faster during times when boats are present, it would imply that the tour boats do not significantly disturb the water more than natural variability does. Aided by onshore cameras and commercial tour boat schedules, the water current profile time series was split into two parts: times when commercial tour boats were present above and in the vicinity of the Sweepstakes, and when they were absent. Additionally, the data set when boats were not present was truncated to only cover the days for which boat presence data was available which is 9:00 EST to 16:00 EST from 16 June 2015 to 3 September 2015. While histogram of the current speeds when boats were present and absent reveal similar speeds and distributions, more frequent high-end speeds appear to occur when boats are absent at the ADP (prow) and when boats are present at the HR-ADCP (approximately 1 m from the port side of the wreck) (Fig 10). While both mean and 90$^{th}$ percentile water currents measured at the side of the shipwreck (AWAC and HR-ADCP) were faster when boats were present, they were slower at the prow (ADP), supporting the visual analysis of the histograms (Table 3).

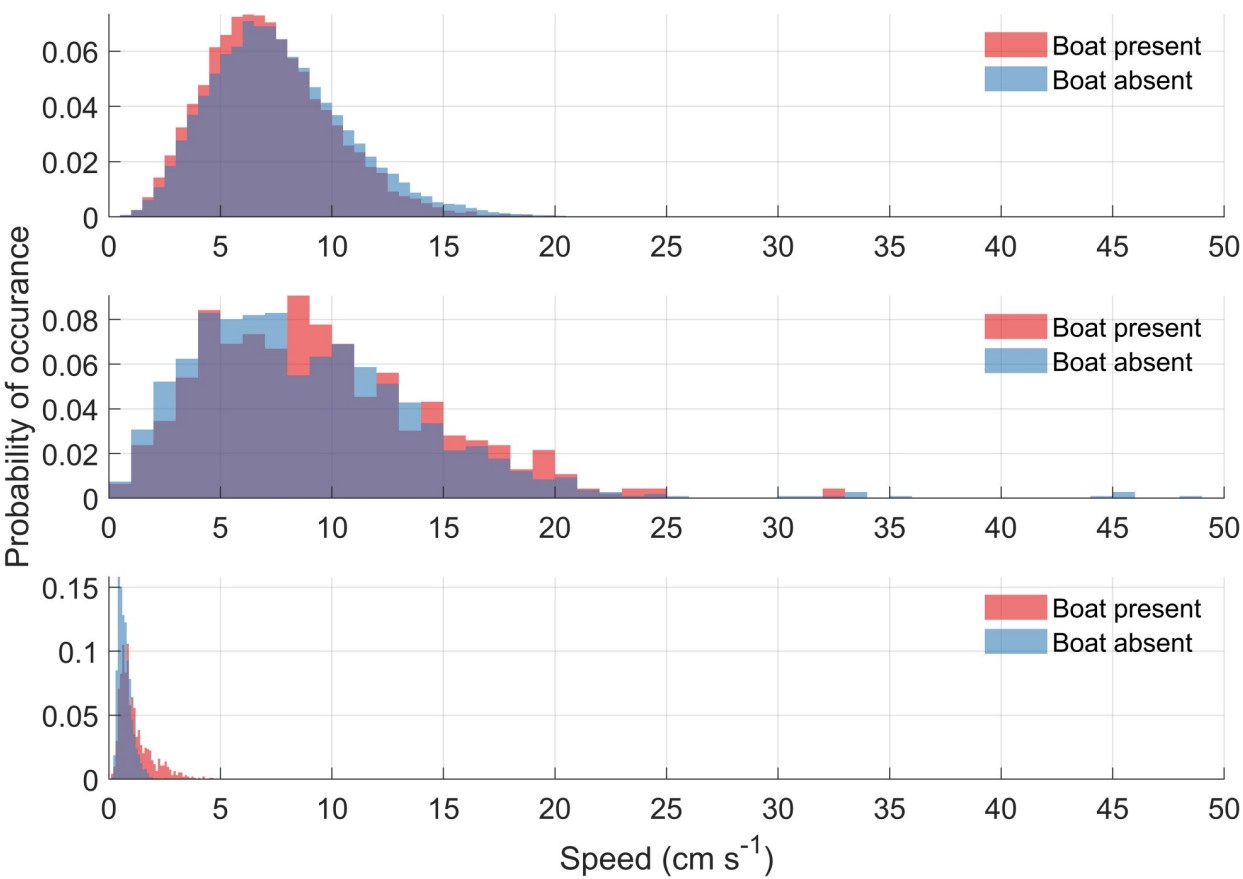

**Fig 10. Histogram analyses of currents with the presence and absence of boats.** (a) ADP located at the prow (b) AWAC located at the side of the shipwreck, and (c) HR-ADCP located at the side of the shipwreck. Normalized frequency on the y-axis is the probability of occurrences which varies between 0–1. The blue color denotes the velocities at the times that the tour boats were absent while red color represents the water currents at the times that the boats were present. The tour boats were allowed from 9:00 EST to 16:00 EST from 16 June 2015 to 3 September 2015. In almost all cases water velocities are less than predicted erosion threshold of 27 cm s$^{-1}$, the fastest observed currents are when boats were absent in (a) and (b).

## Effect of tour boats on water pressure

Tour boats could potentially increase the dynamic pressure upon the shipwreck, so it is important to determine natural changes due to waves and surface seiches, compared to that caused by propeller wash. A pressure sensor attached to the bottom-mounted, HR-ADCP was

**Table 3. Bottom (1.5 m from bed) water current speed statistics during boat presence and absence periods in the vicinity of the Sweepstakes.**

|  | ADP | | AWAC | | HR-ADCP | |
|---|---|---|---|---|---|---|
|  | Boat present | Boat absent | Boat present | Boat absent | Boat present | Boat absent |
| 5th percentile (cm s$^{-1}$) | 1.8 | 1.9 | 2.7 | 2.4 | 0.4 | 0.4 |
| 25th percentile (cm s$^{-1}$) | 4.3 | 4.5 | 5.5 | 5.3 | 0.7 | 0.5 |
| 75th percentile (cm s$^{-1}$) | 9.6 | 10.1 | 12.5 | 12.2 | 1.4 | 0.9 |
| 90th percentile (cm s$^{-1}$) | 11.0 | 11.8 | 16.6 | 15.3 | 2.3 | 1.2 |
| 95th percentile (cm s$^{-1}$) | 14.4 | 15.6 | 19.2 | 18.9 | 2.7 | 1.3 |
| Mean Speed (cm s$^{-1}$) | 7.2 | 7.6 | 9.4 | 8.7 | 1.2 | 0.7 |
| Std. dev. (cm s$^{-1}$) | 2.9 | 3.1 | 4.9 | 4.7 | 0.7 | 0.3 |

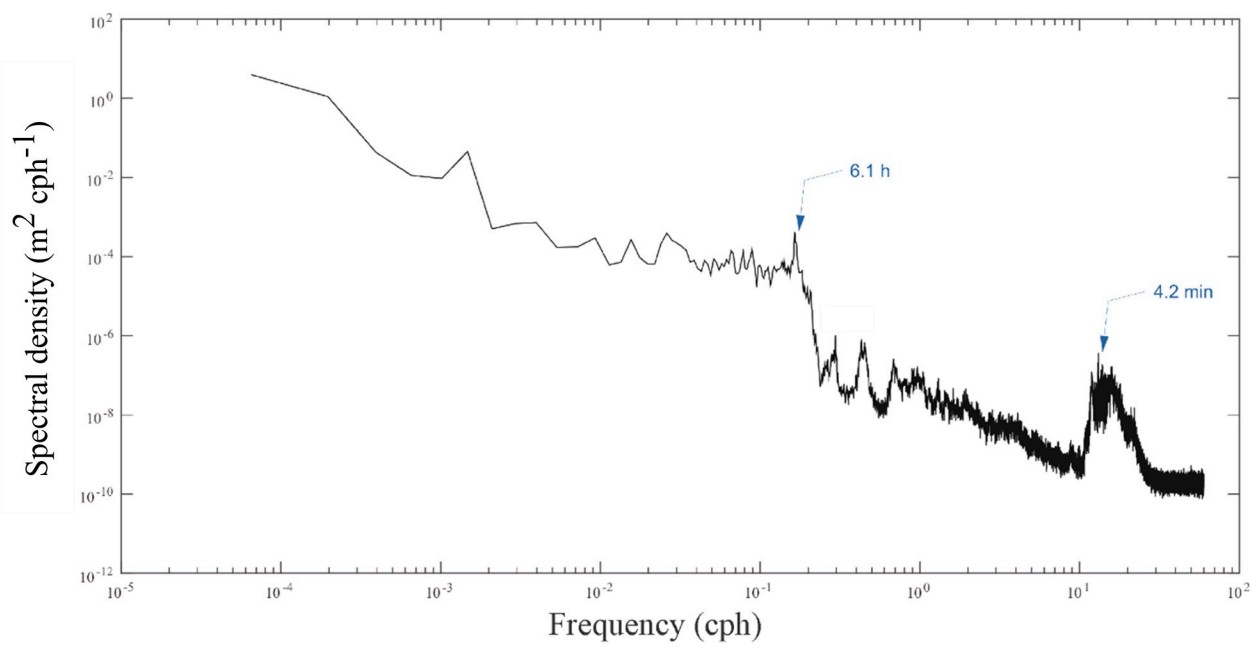

**Fig 11. Spectral density diagram for water pressure perturbations.** Measured at the HR-ADCP on the portside of the wreck (Location 6 in Fig 3). Note that the 4.2-minute period corresponds to the resonant frequency of the harbour.

programmed to record high frequency water pressure and water surface elevation. The spectrum analysis of the pressure perturbation, given in Eq (3), near the Sweepstakes shows a distinct 4.2-minute period (Fig 11) which corresponds well to the harbour's resonant frequency (Eq 2).

The observed pressure perturbation were small suggesting that neither vessels or waves were greatly influencing bottom pressure during the observation period. The maximum amplitude of pressure perturbation near the Sweepstakes was generally greatest during the day, but also experienced peaks during the evenings and nighttime (Fig 12). The maxima in pressure

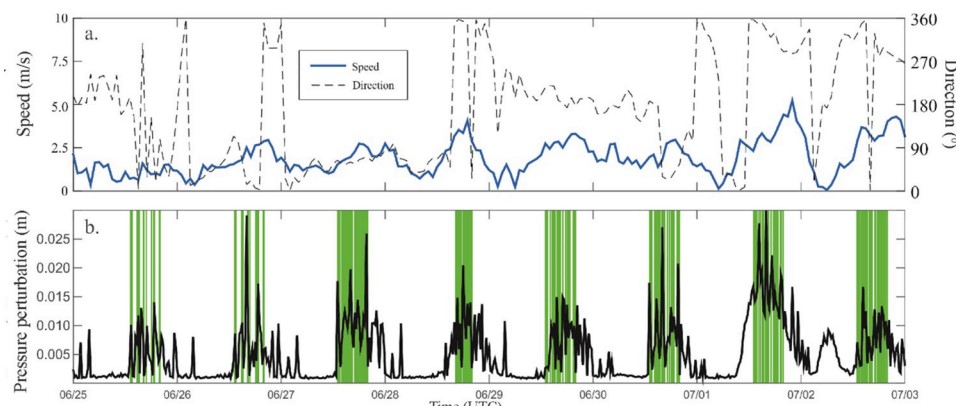

**Fig 12. Comparison of the amplitude of pressure perturbations with wind and the presence of tour boats for 25 June 2015 to 3 July 2015.** (a) Wind velocity and direction (in azimuthal direction, 0 is north) at Tobermory Airport Weather Station (Hourly Data). (b) Maximum of observed pressure perturbations (obtained using high-pass filtered at 4 minutes). In panel (b), the green background areas indicate the presence of tour boats (based on the camera recorded data).

fluctuations generally correspond with peaks in the wind speed data (Fig 12A) and the presence of tour boats (green bands in Fig 12B) during the day. Evening and nighttime peaks generally correspond to peaks in wind speed, with a notable exception occurring the early morning of 2 July (note that the peak around 06:00 UTC corresponds to 02:00 EDT). Boat access to the Sweepstakes was restricted to 09:00–16:00 from 26 June onward, except for Sundays, when they were permitted from 12:00–16:00. Diving was permitted on Sundays from 09:00–12:00, and from 16:00–22:30 every other day from 26 June onwards. Before 26 June there were no restrictions, with boats and divers sharing the site.

## Video observations of biological activities around the shipwreck

A careful analysis of the 100+ hours underwater video camera footage revealed that there were no significant sediment resuspension events during the field experiment (See highlights at https://youtu.be/3i0ORJ_EUS4), which is consistent with the very low currents measured during this time. A byproduct of watching the video was that the fish activity was typically seen to be greatest in the evening hours (after approximately 17:00 EST) and was consistently less in the morning and afternoon hours. Further, round gobies were by far the most numerous fish species present, being almost ubiquitous. Round gobies are known to eat native benthic fishes such as sculpins and darters that dwell on the harbour bed, such that they can cause some bioturbation and sediment resuspension. An average of 10.1 and 14.5 goby fish were observed (in the camera window) per second when tour boats were present and absent, respectively. While this could be due to fish avoiding tour boats, it could also be that fish activity naturally increases in the evenings, which coincides with times when boats are not present. Other species observed to a lesser extent include lake trout (*Salvelinus namaycush*), freshwater drum (*Aplodinotus grunniens*), common carp (*Cyprinus carpio*), common shiner (*Luxilus cornutus*), brook stickleback (*Culaea inconstans*) and double-crested cormorant (*Nannopterum auritum*).

## Discussion

For over 130 years the hull of the Sweepstakes has rested upright, nearly intact, at the head of Big Tub Harbour. With time, as the wood decomposes and metal corrodes, the vulnerability of the wreck to collapse and further deterioration increases. Understanding the nature and source of the forces that could potentially impact the integrity of the site helps to inform and guide possible management actions. Hence, we investigated the summer visitor season water movements around the Sweepstakes to differentiate and quantify the effect of natural and human-derived water movements using spatial and temporal observations of temperatures and currents. Currents flowing around underwater shipwrecks increase flow velocity and turbulence intensity, such that resulting scouring can ultimately lead to failure and collapse of the structure [19]. Scouring around the Sweepstakes can be attributed to one or a combination of wind-driven currents, gravity flows due to upwelling events, surface wave orbital velocities, and flows induced by the wakes of tour boats or skin divers. To investigate which, if any of these processes are responsible for the observed scour around the Sweepstakes, we studied the individual forcings using high frequency temperature and current observed at the immediate vicinity of the shipwreck and at the mouth of Big Tub Harbour. Our main conclusion is that the current management approach, that limits the time and speed that tour boats can visit the site, results in sufficiently weak currents such that there is low potential for sediment erosion. This helps to inform and support on-going management approaches. The site is very protected from currents originating in Georgian Bay, and the natural currents here are generally weak in summertime.

Previous studies in Fathom Five National Marine Park found that vigorous movements of the thermocline [20], but due to protected geometry of big tub harbour, the upwelling of cold water is fairly passive near the Sweepstakes. Our field temperature observations show gradual warming of the entire water column, reaching a maximum of 20˚C. This is persistent throughout the water column and found everywhere in Big Tub Harbour. Due to wind setup in Lake Huron and Georgian Bay, internal waves can form at the thermocline and propagate through the lake. When the amplitude of these internal waves is large enough, they can propagate into Big Tub Harbour. The resulting internal waves are identified as episodic upwelling events in the temperature records (Fig 4). As the internal wave runs up the harbour bed, shoaling and wave breaking could potentially occur, imparting energy and turbulence into the system which could assist in re-suspension of bottom sediment [36, 37]. These upwelling events can clearly be seen in the temperature data sets for all the thermistors located in Big Tub Harbour (Figs 4 and 6). However, these currents are slow and their signature was more difficult to detect in the current velocity records. There was no discernable difference in mean current speeds at the AWAC and HR-ADCP, while they were 0.2 cm s$^{-1}$ faster during non-upwelling events at the ADP (Table 2). More importantly, the difference in current speeds at the 90$^{th}$ percentile between upwelling and non-upwelling events only varied between 0.1–0.4 cm s$^{-1}$ across all three current meters, with no consensus as to which period experienced faster currents (Table 2). This suggests gravity-driven currents do not induce substantial current speeds in the vicinity of the Sweepstakes, and that the internal gravity flows induced by the cold-water intrusions did not substantially influence sediment resuspension.

Surface seiches play a role similar to tides in large lakes and drive noticeable changes in water depth, as well as driving currents in and out of Big Tub harbour. Field observations of currents found that depth averaged motions near the bottom (i.e. 1.5 m from the harbour bed) with the significant peaks at 12 and 24 hr periods. Thus, to study the currents' variability at the bottom near Sweepstakes, we use depth-averaged speeds. The analysis of depth-averaged bottom currents indicated mean speeds of 10 cm s$^{-1}$ s at the prow (ADP) of the shipwreck, 9 cm s$^{-1}$ several meters off the port side (AWCA), and less than 1 cm s$^{-1}$ speeds directly at the port side (HR-ADCP) of the shipwreck (Fig 9). The increase in flow velocity at the prow is likely due to the conservation of mass as the water masses flow in and out of the harbour [19]. Similarly, the much lower velocities directly at the open harbour side of the structure could represent a stagnation point of water flowing around the wreck, as visualized schematically in the streamline diagram (Fig 13), reminiscent of work done by Quinn [19]. The obvious difference to the work on Quinn [19] is that the currents induced by seiches at this protected site are a fraction of those driven by dynamic ocean tides.

The major motivation for this study was to determine if any observed water currents could drive sediment suspension, so we now discuss the currents that would needed to drive erosion. Particle size analysis conducted by Boyce [18] suggests the bottom sediment in the vicinity of the Sweepstakes, but outside of the scouring, was predominantly between 62–176 microns (~95% sand, with silt and clay making up the remainder). Within the scour ring, sediment grain size was markedly finer on the inland (west) side of the wreck (typically 31–125 microns with a mean of 74 microns, composed of ~64% sand, ~34% silt, and ~2% clay) than on the open harbour (east) side (typically 125–250 microns with a mean of 187 microns, composed of ~97% sand, ~2% silt, and clay and ~1% gravel) [18]. The relative scarcity of sediments with diameter less than 125 microns in the scour ring on the open-harbour side of the wreck suggests scour and erosion is occurring through suspended load transport of the very fine-grained particles [31]. Current velocities on the order of 27 cm s$^{-1}$ at 1.5 m above the harbour bed are required to initiate suspended load transport of the very fine-grained sediment in the vicinity of the Sweepstakes (as discussed in Methods section). With respect to our observations, we see

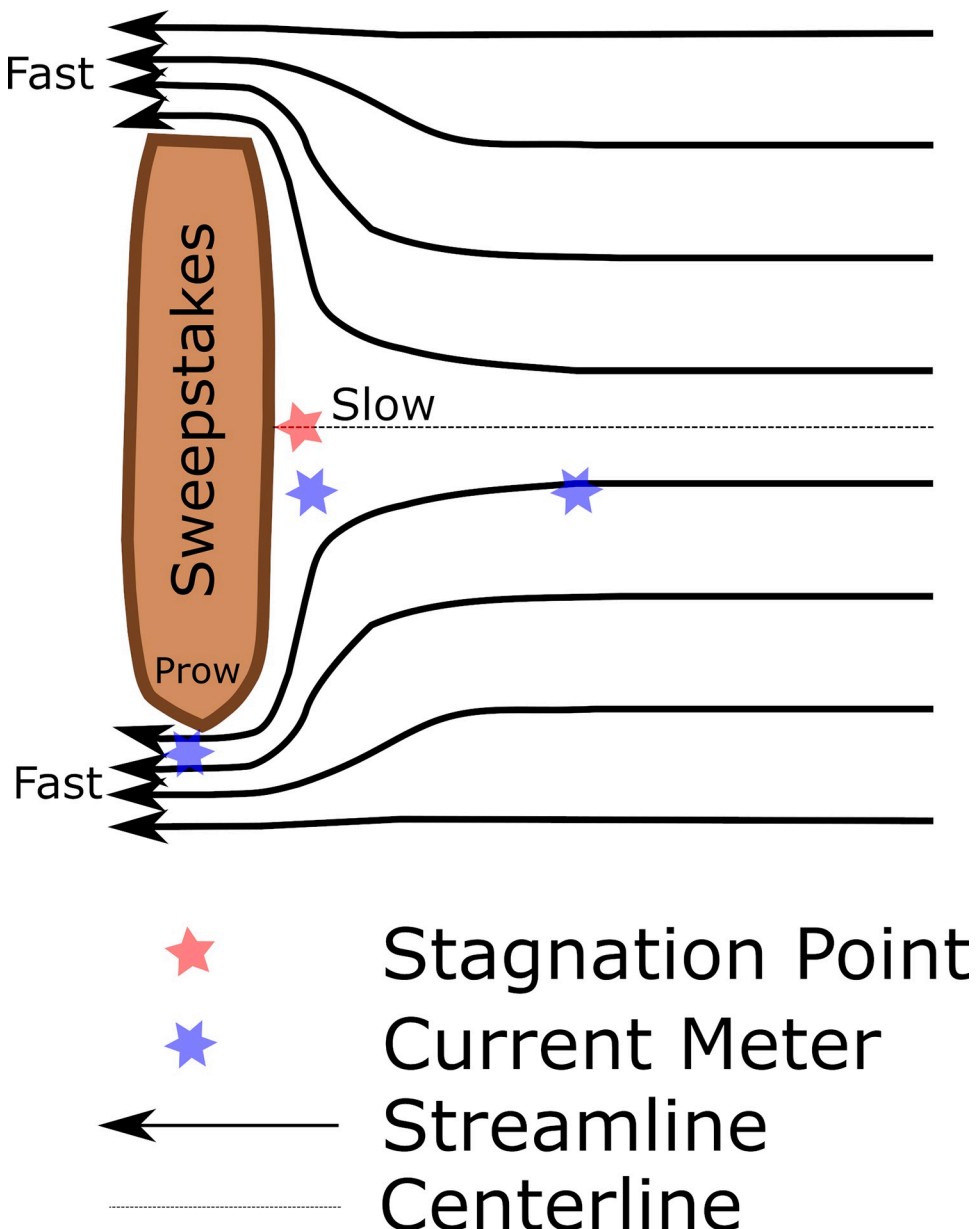

**Fig 13. Idealized streamline diagram of benthic water currents around the Sweepstakes.** Red five-pointed star estimates the stagnation point, where flow is zero. Blue six-pointed stars indicate location of the current meters in relation to the shipwreck. Solid lines with arrows are representative streamlines. Distance between streamlines represents relative flow velocity, with tighter spacing indicating faster flows. Dotted line represents the centerline. Stagnation point lines on centerline for symmetrical obstruction.

that the mean currents were approximately 9 cm s$^{-1}$ and only briefly exceeded the resuspension velocity of 27 cm s$^{-1}$ (see methods section for details) only twice over the 46-day record (out of 3264 observations). Therefore, during instrument deployment, bottom currents driven by both natural and anthropogenic forces were unlikely to result in significant sediment transport. This is corroborated in the underwater video record, which did not observe any resuspension in over 100 h of video footage.

Our main conclusion is that propeller wash was weak, and never higher than the (weak) natural currents present in Big Tub Harbour. Analysis of bottom currents that might be potentially induced by propeller wash (Fig 10) showed that differences in mean currents were small, between 0.4–0.7 cm s$^{-1}$ across the three current meters, and, while they were faster at the side of the wreck when tour boats were present, they were slower at the prow (Table 3). These trends were consistent when analyzing the 90$^{th}$ percentile of current speeds, where differences between boat present and absent periods was between 0.8–1.3 cm s$^{-1}$ (Table 3). This suggests that the propeller wash induced currents are not responsible for the increased intensity of turbulence and scouring around the shipwreck. In addition, the differences in current speeds between boat presence and absence periods, and upwelling and non-upwelling periods, were small with no clear consensus as to which periods were faster suggesting a negligible difference in water currents due to tour boats and upwelling. Despite a notable increase in vessel size (~40% larger) and frequency of use, the current study found that the maximum velocities on the port side of wreck (same observation location as Boyce [18]) were at most ~5 cm s$^{-1.}$ They were always much less than both the maximum values of 12 cm s$^{-1}$ observed by Boyce [18] when tour vessels were operated aggressively, and the isolated peak currents of 11.2 cm s$^{-1}$ and 17 cm s$^{-1}$ Boyce [18] observed in 1993 and 1994, respectively. Interestingly, the fastest velocities were always seen at the prow, and were actually highest when no boats were present. Furthermore, observed currents at all three current meters were less than the critical velocity required for resuspension and no resuspension events were observed during the 2015 summer observation campaign.

Investigations of shallow-water shipwrecks in marine settings with energetic tidal systems have shown that the accretion and erosion processes that lead to scour rely on a complex interplay of wave-induced currents and steady flow, seasonal variability in wave energy, and site conditions [19, 38]. Our findings suggest that these same processes likely influence scour around shipwrecks in lower-energy freshwater lake systems.

Despite a lack of evidence of bottom currents sufficient to initiate sediment movement during the summer of 2015, scour and sediment transport are occurring in the vicinity of the Sweepstakes, as evidenced by the scour ring around the wreck (Figs 1C and 3B). Work done by Skafel (see appendix in [18]) suggested that waves would have to be 0.3 m high to cause erosion. Such waves were never seen in summer (either by us or Boyce), but are very possible during fierce winter storms. While Boyce [18] sometimes observed waves 0.3–0.6 m high at lake entrance to Big Tub Harbour, waves were rarely higher than 0.15 m in the harbour. Furthermore, these large waves were observed in late September and October (i.e., during the fall). It is therefore likely that erosion occurs during the fall and winter, when wave energy and associated current velocities are much higher. While winter storms are not an anthropogenic stressor and therefore not subject to management intervention, future research should investigate natural water movements during the autumn and winter seasons to assess erosional and scour potential outside of the summer tourist season. Multitemporal remote sensing and digital evaluation modeling following Cham et al. [38] could also be applied to investigate the temporal evolution of the scour around the Sweepstakes, both intra- and inter-seasonally, providing details on degradation rates to inform management decisions.

## Conclusion

Quantifying and differentiating natural and human-derived water movements around the wreck of the Sweepstakes is important for informing and guiding management actions. Our extensive field observations during the peak summer visitor season suggest that there does not

appear to be a meaningful difference in water currents around the Sweepstakes between times when tour boats are present or absent, and that in general water currents are weak. This is favourable to the long-term preservation of the Sweepstake shipwreck–one of the most iconic shipwrecks in Canada. Moderate 90th percentile current speeds, small differences between boat present and absent periods, and lack of consensus as to which period experiences faster currents suggest tour boats are unlikely to increase scour potential in the vicinity of the Sweepstakes under the current operation regime. Although there are noticeable pressure perturbations during boat presence periods, likely generated by boat propeller and jet-drive wash, our observations suggest that the averaged current variabilities were insignificant near the harbour bed. Furthermore, the assessment of videos indicate that the boat induced pressure perturbations, instantaneous velocities, and induced dynamic forces on the shipwreck are unlikely to drive resuspension. Based on our observations, the natural and human-derived water movements during the peak visitor season during the summer of 2015 are unlikely to be the cause of the observed scour around the Sweepstakes. Rather the scour is probably caused by winter storms, which should be a focus of future studies. As the Sweepstake site is a restricted area, vessel access is limited to scheduled times and while maneuvering over the site, vessels are required to use the minimum amount of power necessary to maintain safety and to minimize propeller wash. In this context, where human-derived forces do not appear to be significantly different than natural forces, management efforts can continue to focus on monitoring and conserving the shipwreck generally. If, however, there is a change in the context, such as larger vessels or increased propeller/jet-drive wash, then a reexamination of the relative impacts is recommended.

## Supporting information

**S1 Video. Underwater video showing location of ADP at the prow of the Sweepstakes, and the HR ADCP on the portside.**
(MP4)

**S2 Video. Underwater video showing location of AWAC and HR ADCP, along portside of the Sweepstakes.**
(MP4)

**S3 Video. Underwater video showing instruments on Float 3.**
(MP4)

**S4 Video. Underwater video showing instruments on Float 4, and location of Float 3, Float 3, and HR ADCP.**
(MP4)

**S5 Video. Select video clips of aquatic activity from downward-facing underwater camera mounted on ADP frame at the prow of the Sweepstakes.**
(MP4)

## Acknowledgments

We thank Environment and Climate Change Canada and the captains and crews of the Sauger and the Stickleback for logistical support with the field measurements. Scott thanks Bruce Gray, Dive Operations Officer for ECCC (Environment and Climate Change Canada) and his team, and Katrina Keeshig, Parks Canada.

## Author Contributions

**Conceptualization:** Bryan Flood, Mathew G. Wells, Reza Valipour, Scott Parker.

**Data curation:** Bryan Flood, Lakshika Girihagama.

**Formal analysis:** Bryan Flood, Lakshika Girihagama, Reza Valipour, Patricia Semcesen.

**Funding acquisition:** Mathew G. Wells.

**Investigation:** Bryan Flood, Lakshika Girihagama, Reza Valipour, Scott Parker.

**Methodology:** Bryan Flood, Mathew G. Wells, Reza Valipour.

**Project administration:** Mathew G. Wells, Scott Parker.

**Resources:** Mathew G. Wells, Reza Valipour, Scott Parker.

**Supervision:** Mathew G. Wells.

**Visualization:** Bryan Flood, Lakshika Girihagama.

**Writing – original draft:** Bryan Flood, Lakshika Girihagama.

**Writing – review & editing:** Bryan Flood, Lakshika Girihagama, Mathew G. Wells, Reza Valipour, Scott Parker.

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
