## [Decision Letter · Decision Letter 0]

23 Oct 2021

PONE-D-21-24254

Investigating the water movements around a shallow shipwreck in Big Tub Harbour of Lake Huron: implications for managing underwater shipwrecks.

PLOS ONE

Dear Dr. Flood,

Thank you for submitting your manuscript to PLOS ONE. After careful consideration, we feel that it has merit but does not fully meet PLOS ONE’s publication criteria as it currently stands. Therefore, we invite you to submit a revised version of the manuscript that addresses the points raised during the review process.

We look forward to receiving your revised manuscript.

Kind regards,

Haibin Lv

Academic Editor

PLOS ONE

2. We note that Figures 1-3  in your submission contain [map/satellite] images which may be copyrighted. All PLOS content is published under the Creative Commons Attribution License (CC BY 4.0), which means that the manuscript, images, and Supporting Information files will be freely available online, and any third party is permitted to access, download, copy, distribute, and use these materials in any way, even commercially, with proper attribution. For these reasons, we cannot publish previously copyrighted maps or satellite images created using proprietary data, such as Google software (Google Maps, Street View, and Earth). For more information, see our copyright guidelines: http://journals.plos.org/plosone/s/licenses-and-copyright.

a. You may seek permission from the original copyright holder of Figures 1-3 to publish the content specifically under the CC BY 4.0 license. 

3. Please include your tables as part of your main manuscript and remove the individual files. Please note that supplementary tables (should remain/ be uploaded) as separate ""supporting information"" files"""

Reviewers' comments:

Reviewer's Responses to Questions

**Comments to the Author**

1. Is the manuscript technically sound, and do the data support the conclusions?

Reviewer #1: Partly

Reviewer #2: Yes

2. Has the statistical analysis been performed appropriately and rigorously? 

Reviewer #1: N/A

Reviewer #2: No

3. Have the authors made all data underlying the findings in their manuscript fully available?

Reviewer #1: Yes

Reviewer #2: Yes

4. Is the manuscript presented in an intelligible fashion and written in standard English?

Reviewer #1: Yes

Reviewer #2: Yes

5. Review Comments to the Author

Reviewer #1: - We prefer if you use the third person singular, instead of the first person singular or plural (e.g. "we").

- More suitable title should be selected for the article.

- The abstract should state briefly the purpose of the research, the principal results and major conclusions. An abstract is often presented separately from the article, so it must be able to stand alone.

- It is suggested to present the structure of the article at the end of the introduction.

- The major defect of this study is the debate or Argument is not clear stated in the introduction session. Hence, the contribution is weak in this manuscript. I would suggest the author to enhance your theoretical discussion and arrives your debate or argument.

- More suitable title should be selected for the table 1 instead of “Summery of instruments deployed in Big Tub Harbour in 2015.”.

- It is suggested to compare the results of the present research with some similar studies which is done before.

- It is suggested to add articles entitled “Nazarnia et al. A Systematic Review of Civil and Environmental Infrastructures for Coastal Adaptation to Sea Level Rise”, “Gholami & Baharlouii. Monitoring Long-term Mangrove Shoreline Changes along the Northern Coasts of the Persian Gulf and the Oman Sea” and “Cham et al. An Analysis of Shoreline Changes Using Combined Multitemporal Remote Sensing and Digital Evaluation Model” to the literature review.

- Page 14: the following paragraph is unclear, so please reorganize that:

“We compare the temperature time series acquired in the direct vicinity of the Sweepstakes and at the mouth of the harbour to see if there are any intrusive cold gravity flows in Big Tub Harbour induced by the upwelling of cold waters in Lake Huron. The upwelling events are identified as a drop in water temperature by 5-8 0C in the space of few hours.”

- Much more explanations and interpretations must be added for the Results, which are not enough.

- Please make sure your conclusions' section underscore the scientific value added of your paper, and/or the applicability of your findings/results, as indicated previously. Please revise your conclusion part into more details. Basically, you should enhance your contributions, limitations, underscore the scientific value added of your paper, and/or the applicability of your findings/results and future study in this session.

- DOI of the references must be added (you can use “" ext-link-type="uri" xlink:type="simple">https://crossref.org/").

Reviewer #2: The authors collected a wide variety of field measurements at multiple locations and times and aimed to assess the potential impact of human-derived water movements on shipwrecks. They quantify various parameters of human-derived currents and of natural water currents. It is quite laborious to well collect and analyze such a rich data, while the logical structure of the context is clear. In many instances, the tourism developments impact the conservation and preservation of the natural environment and cultural heritages. Scientific evidence and surveillance regarding to the damaging effect would conduce to develop their mutually beneficial module. This study gave real maneuvers on acquiring observation and objective inference in a scientific way. To facilitate practical reference, some suggestions proposed. Although the results driven from various measurements were summarized in tables and elaborated in content, no inferential statistical analyses were performed. According to the description on data collection, repeated measurement analysis or time series analysis could be used. Other comparable methods are also usable, as long as their applicability is reasonable. At least, 95% confidence intervals for point estimates, e.g., in tables, are recommended to be supplemented. In addition, condensed statements regarding the limitations mentioned in discussion and conclusions should be added in abstract.

6. PLOS authors have the option to publish the peer review history of their article (what does this mean?). If published, this will include your full peer review and any attached files.

Reviewer #1: No

Reviewer #2: No

---

## [Author Response · Author response to Decision Letter 0]

11 Apr 2022

Each comment from the reviewers and editors has been considered, and a response provided in the document "Response to Reviewers."

---

## [Decision Letter · Decision Letter 1]

18 Oct 2022

PONE-D-21-24254R1Investigating water movements around a shallow shipwreck in Big Tub Harbour of Lake Huron: implications for managing and preserving underwater shipwrecksPLOS ONE

Dear Dr. Flood,

Thank you for submitting your manuscript to PLOS ONE. After careful consideration, we feel that it has merit but does not fully meet PLOS ONE’s publication criteria as it currently stands. Therefore, we invite you to submit a revised version of the manuscript that addresses the points raised during the review process.

We look forward to receiving your revised manuscript.

Kind regards,

Ram Kumar

Academic Editor

PLOS ONE

Journal Requirements:

Additional Editor Comments:

Thanks for submitting the manuscript to PloseOne . All the reviewers found manuscript useful. Some minor improvements are required to enhance readability and application. I have perused the Ms. and found the study s been done with proper objectives and definitive results are presented. A sentence on application and understanding of water movements in other similar ecosystems like Big Tub Harbour of Lake Huron would further benefit the paper.

Thanks

Reviewers' comments:

Reviewer's Responses to Questions

**Comments to the Author**

1. If the authors have adequately addressed your comments raised in a previous round of review and you feel that this manuscript is now acceptable for publication, you may indicate that here to bypass the “Comments to the Author” section, enter your conflict of interest statement in the “Confidential to Editor” section, and submit your "Accept" recommendation.

Reviewer #2: All comments have been addressed

Reviewer #3: All comments have been addressed

Reviewer #4: (No Response)

Reviewer #5: (No Response)

2. Is the manuscript technically sound, and do the data support the conclusions?

Reviewer #2: Yes

Reviewer #3: Yes

Reviewer #4: Yes

Reviewer #5: Yes

3. Has the statistical analysis been performed appropriately and rigorously? 

Reviewer #2: Yes

Reviewer #3: Yes

Reviewer #4: No

Reviewer #5: Yes

4. Have the authors made all data underlying the findings in their manuscript fully available?

Reviewer #2: Yes

Reviewer #3: Yes

Reviewer #4: Yes

Reviewer #5: Yes

5. Is the manuscript presented in an intelligible fashion and written in standard English?

Reviewer #2: Yes

Reviewer #3: Yes

Reviewer #4: Yes

Reviewer #5: Yes

6. Review Comments to the Author

Reviewer #2: The author did appropriate revision partially and provided explanations on revisions which did not comply to the modification suggestions. This study did contribute abundant evidence and demonstrated an objective assessment maneuver for underwater antiquities deteriorated by human activities. I accept the authors' way and their explanations on revisions.

Reviewer #3: I congratulate the authors for considering and efficiently revising the manuscript. The manuscript now seems to be of publishable form.

Reviewer #4: A few of my concerns are as follows:

1. Justification or interpretation should be limited to the discussion section, the glimpses of which could be seen in the results sections.

2. Same goes for the conclusion , which has some results mentioned in it, instead of only concluding statements.

3. Line no: 138 "affect" needs to change with "effect".

4. Line no: 725-728 needs improvement. The statement repeats the citation of Boyce, three times in a single sentence, while making it unclear if the data mentioned, belongs to Boyce, or your study, specifically because of the use of "our study" in between the sentence.

Reviewer #5: It is commendable that the authors collected a range of field measurements in various locations throughout the study site. The hypothesis for this journal also seems to make sense. I appreciate the authors' efforts; however, I would want to make a few suggestions.

• The authors have not been precise; for example, lines 547 to 552 contain numerous repetitions of the same statement.

• Also, I would like recommend some more minor adjustments like: -

i. Line 41 (keyword section)- There are two phrases here, namely "shallow shipwrecks" and "water movement"

these phrases may be substituted with alternative phrases as they are already present in line 1, i.e., title section.

ii. Line 100- “A ring of erosion is visible around the Sweepstake where no aquatic vegetation (Chara sp.)”, why is

Chara sp. listed in brackets while it is said that there is no vegetation present?

iii. Line 105- full form/detailed about ADP should be mentioned.

iv. Line 427 to 428- “Stratified water bodies often have strong currents associated with internal waves that can be a

strong source of oscillating currents” any citations would strengthen this assertion.

v. Line 465- “event or period” specific phrase can be maintained.

vi. Line 469- “variability or maximum” specific phrase can be maintained.

7. PLOS authors have the option to publish the peer review history of their article (what does this mean?). If published, this will include your full peer review and any attached files.

Reviewer #2: No

Reviewer #3: No

Reviewer #4: **Yes: **DEVESH KUMAR YADAV

Reviewer #5: **Yes: **Abhishek Patel

---

## [Author Response · Author response to Decision Letter 1]

4 Dec 2022

Please note that the response to reviewers is contained in the submitted document "Response to Reveiwers.doc"

---

## [Editor Report · Decision Letter 2]

19 Dec 2022

Investigating water movements around a shallow shipwreck in Big Tub Harbour of Lake Huron: implications for managing and preserving underwater shipwrecks

PONE-D-21-24254R2

Dear Dr. Bryan Flood,

We’re pleased to inform you that your manuscript has been judged scientifically suitable for publication and will be formally accepted for publication once it meets all outstanding technical requirements.

Kind regards,

Ram Kumar, Ph.D.

Academic Editor

PLOS ONE
---

## [Editor Report · Acceptance letter]

22 Dec 2022

PONE-D-21-24254R2 

Investigating water movements around a shallow shipwreck in Big Tub Harbour of Lake Huron: implications for managing and preserving underwater shipwrecks 

Dear Dr. Flood:

I'm pleased to inform you that your manuscript has been deemed suitable for publication in PLOS ONE. Congratulations! Your manuscript is now with our production department. 

Kind regards, 

on behalf of

Professor Ram Kumar 

Academic Editor

PLOS ONE